# Hybrid Nano Flake-like Vanadium Diselenide Combined on Multi-Walled Carbon Nanotube as a Binder-Free Electrode for Sodium-Ion Batteries

**DOI:** 10.3390/ma16031253

**Published:** 2023-02-01

**Authors:** Youngho Jin, Min Eui Lee, Geongil Kim, Honggyu Seong, Wonbin Nam, Sung Kuk Kim, Joon Ha Moon, Jaewon Choi

**Affiliations:** 1Department of Chemistry and Research Institute of Natural Science, Gyeongsang National University, Jinju 52828, Republic of Korea; 2Energy & Environment Laboratory, KEPCO Research Institute, Daejeon 34056, Republic of Korea

**Keywords:** metal chalcogenides, nano-flake, vanadium diselenide, MWCNT, binder free, sodium-ion batteries

## Abstract

As the market for electric vehicles and portable electronic devices continues to grow rapidly, sodium-ion batteries (SIBs) have emerged as energy storage systems to replace lithium-ion batteries (LIBs). However, sodium-ion is heavier and larger than lithium-ion, resulting in volume expansion and slower ion transfer. It is necessary to find suitable anode materials with high capacity and stability. In addition, wearable electronics are starting to be commercialized, requiring a binder-free electrode used in flexible batteries. In this work, we synthesized nano flake-like VSe_2_ using organic precursor and combined it with MWCNT as carbonaceous material. VSe_2_@MWCNT was mixed homogenously using sonication and fabricated film electrodes without a binder and substrate via vacuum filter. The hybrid electrode exhibited high-rate capability and stable cycling performance with a discharge capacity of 469.1 mAhg^−1^ after 200 cycles. Furthermore, VSe_2_@MWCNT exhibited coulombic efficiency of ~99.7%, indicating good cycle stability. Additionally, VSe_2_@MWCNT showed a predominant 85.5% of capacitive contribution at a scan rate of 1 mVs^−1^ in sodiation/desodiation process. These results showed that VSe_2_@MWCNT is a suitable anode material for flexible SIBs.

## 1. Introduction

Sodium-ion batteries (SIBs) are garnering attraction as a next-generation energy storage device to replace lithium-ion batteries (LIBs) thanks to their resource-abundant and price-effective advantages [1,2]. SIBs have a charging protocol similar to LIBs. Their disadvantage is that sodium ions have a larger radius than lithium ions [3,4]. It is necessary to find an appropriate anode material for SIBs in terms of energy density, specific capacity, and stability. Among various anode material candidates (carbon, metal oxide, metal sulfide, organic materials, etc.) [5,6,7,8], metal selenides have a high capacity and a lot of active sites. However, metal selenides have problems due to their low cycle stability and electrical conductivity [9,10]. These problems are mitigated through compounding with carbonaceous materials such as reduced graphene oxide (rGO) and carbon nanotube (CNT) [11,12,13]. Usually, when preparing an anode electrode using the slurry-casting method, the slurry is coated on the copper foil using a binder such as a polyvinylidene fluoride (PVDF), polytetrafluoroethylene (PTFE), sodium carboxymethyl cellulose (CMC), polyvinyl alcohol (PVA), or styrene butadiene rubber (SBR). However, since binders are generally electrochemically inactive and insulating materials, an experimental process using binders may reduce electrical conductivity and cause a side reaction with the electrolyte [14]. Moreover, most binders are unstable at high temperatures exceeding 200 °C and reduce the overall energy density of batteries by increasing the weight and volume of electrodes [15,16,17,18]. With technological advances, the demand for flexible and wearable batteries has surged, especially in relation to binder-free electrodes [19]. In this paper, we made CNTs (carbon nanotubes) composite binder-free film electrodes to improve conductivity through a vacuum filter. CNTs are widely investigated in various fields for their chemical stability, electrical conductivity, and large surface area, and are used as energy storage, electronics, etc. [20,21]. Research papers on improved characteristics of CNTs have been analyzed as functionalized covalent or non-covalent [22]. Recently, the covalent functionalized CNTs composite with metal chalcogenide is starting to be applied in anode for alkali ion batteries. For example, T. Hou et al. reported ZnS/CNT composite through MWCNT processed acid treatment. The ZnS/CNT electrodes exhibit long-term cycle stability and a capacity of 333 mAhg^−^^1^ at 2 Ag^−^^1^ over 4000 cycles for LIBs and 314 mAhg^−^^1^ at 5 Ag^−^^1^ after 500 cycles for SIBs [23]. However, the disorganized conjugation system of CNT and covalent functionalized CNTs are not suitable for application for high conductivity [24]. On the contrary, non-covalent (π-π interaction, Van der Waals force, electrostatic, etc.) functionalized CNTs have delocalized the π- electron through π-stacking or/and Van der Waals force and have a high level of electron conductivity [24]. Thus, the binder-free electrode can be made by combining metal selenides and CNT, which is a carbon material with excellent flexibility and conductivity, through vacuum filtering. For example, M. Chen et al. fabricated a binder-free anode using a 2D ultrathin SnO nanoflakes array grown directly on GF/CNTs substrate [25]. Y. Wang et al. synthesized binder-free WS_2_/CNT-rGO aerogel hybrid nanoarchitecture electrodes and showed outstanding electrochemical performance for both LIBs and SIBs [26]. Layered metal chalcogenides such as MoSe_2_, WS_2_, and TiS_2_ have been applied in energy storage due to their thick atomic layers and 2D morphology [27,28,29]. Y. Tang et al. have synthesized carbon-stabilized interlayer-expanded few-layer MoSe_2_@C nanosheets and found that they could exhibit a reversible capability of 421 mAhg^−^^1^ at 0.2 Ag^−^^1^ [30]. Y. Liu et al. showed that WS_2_/NC nanosheets exhibited a reversible specific capacity of 180.1 mAhg^−^^1^ at a current density of 1.0 Ag^−^^1^ after 400 cycles [31]. Vanadium diselenide, which has a typical layered structure metal chalcogenide, has large interlayer spacing (6.11 Å) and has weak Van der Waals force between the layers. Consequently, vanadium diselenide is expected to have great potential as an alternative anode material for sodium ion batteries, in addition to its applications in anodes or cathodes for metal ion batteries [32]. Because of the strong electron coupling between V^4+^-V^4+^ pairs, vanadium diselenide induces metallic properties and can accommodate sodium ion in vanadium diselenide, which shows changes in the valence state from V^+5^ to V^+2^ [33]. Moreover, vanadium diselenide has a crystal structure similar to that of graphite. VSe_2_ shows conversion reactions during charge/discharge processes [34]. These characteristics facilitate the diffusion of large-sized alkali ions. Theoretically, 1 mol of VSe_2_ can hold 4 mol of sodium ions and electrons, which exhibits a high theoretical specific capacity [35]. In this study, nano flake-like vanadium selenides (VSe_2_) were synthesized by a simple colloidal method and hybridized with multi-walled carbon nanotubes (MWCNT) to be used as an anode electrode without binder, conductive carbon, and substrate. VSe_2_@MWCNT nanohybrids showed high-rate performance and long-cycle stability with a discharge capacity of 469.1 mAhg^−^^1^ at a current density of 0.01 Ag^−^^1^ after 200 cycles. Therefore, instead of combined MWCNT, which shows poor cycle stability, this paper suggests alternative anode materials for flexible SIBs.

## 2. Materials and Methods

### 2.1. Synthesis of 1,3-Dimethyl-imidazoline-2-selenone

Synthesis of 1,3-dimethyl-imidazoline-2-selenone was performed with slight modification on the previous report [36,37]. Briefly, Iodomethane (CH_3_I, 2 mL, 16 mmol, 1.3eq. (to 1-Methylimidazole) JUNSEI, Tokyo, Japan) was added to a 200 mL two-neck Schlenk flask. After that, 1-Methylimidazole (C_4_H_6_N_2_, 2 mL, 12 mmol, Alfa Aesar, Seoul, Republic of Korea) was poured with Methanol (MeOH, 30 mL, SAMCHUN, Pyeongtaek, Republic of Korea) into a Schlenk flask and stirred at 800 rpm at room temperature overnight. The supernatant was removed and we collected a white colored product (1,3-dimethylimidazolium iodide, 2.91 g) was collected. Selenium (Se, 3.15 g, 40 mmol (3eq. to 1,3-dimethylimidazolium iodide), Acros organics, Fairlawn, NJ, USA) was then added, followed by addition of potassium carbonate anhydrous (K_2_CO_3_, 9.36 g, 67 mmol, SAMCHUN, Pyeongtaek, Republic of Korea). The reaction was stirred over one day at room temperature. Then, the product was filtrated with Celite and the extra solvent was evaporated. The crude product was extracted using dichloromethane (DCM, SAMCHUN, Pyeongtaek, Republic of Korea) and distilled water (DI water).

### 2.2. Synthesis of Nano Flake-like VSe_2_ and VSe_2_@MWCNT

Nano flakes-like VSe_2_ was synthesized via a wet chemical method with a surfactant for shape control. A 50 mL two-neck Schlenk flask was poured with oleylamine (OAm, 12 mL, technical grade 70%, Sigma-Aldrich, St. Louis, MO, USA) and dried under vacuum conditions at 150 °C for over one hour. After heating oleylamine, vanadium chloride (VCl_3_, 0.1 g, Sigma-Aldrich, St. Louis, MO, USA) was added, followed by the addition of 1,3-dimethyl-imidazoline-2-selenone dissolved in dichloromethane (DCM, 3 mL, SAMCHUM, Pyeongtaek, Republic of Korea) solution under argon condition. The reaction was heated to 200 °C for 3 h. A black-colored product was cooled at room temperature and washed with methanol and hexane. The obtained product was then dried in vacuum conditions.

### 2.3. Material Characterization

Morphologies of the VSe_2_@MWCNT and MWCNT were examined using a scanning electron microscope (SEM, JSM-7601F, JEOL, Tokyo, Japan) and a transmission electron microscope (TEM, FEI RF30ST, Philips, Amsterdam, The Netherlands) equipped with an energy-dispersive spectrometer (EDS, Ultim Max, Oxford Instruments, Abingdon on Thames, UK). X-ray diffraction (XRD) were conducted using a D8 Advance A25 (Bruker, Billerica, MA, USA) at 40 kV and 40 mA to characterize compositions of compounds. Raman spectra were recorded on Renishaw InVia (Renishaw, Wotton-under-Edge, UK) with a wavelength of 514.5 nm. X-ray photoelectron spectroscopy (XPS) was carried out using the Thermo VG scientific Sigma Probe spectrometer (Sigma probe, Thermo VG scientific, East Grinstead, UK) with a monochromatic photon energy of 1486.6 eV (Al Kα). The specific surface area was determined from N_2_ adsorption–desorption isotherms measured using the BELSORP-mini Ⅱ (MicrotracBEL, Osaka, Japan).

### 2.4. Electrochemical Properties Evaluation

VSe_2_@MWCNT and MWCNT electrodes were made without a binder and substrate. The VSe_2_ and MWCNT (VSe_2_/MWCNT 1:1) were added in N,N-dimethylformamide and dispersed via sonication. The well-dispersed mixture was filtered through an Anodisc membrane (47 nm in diameter, 0.2 mm pores, Whatman, Maidstone, UK) and placed in a convection oven to dry. After that, VSe_2_@MWCNT film was separated from the filter and used directly as anode for sodium-ion batteries. Half-coin cells were assembled in a glove box to prevent sodium metal contamination. Half-cion cell (CR2032) was prepared using VSe_2_@MWCNT with the MWCNT electrode as the working electrode, sodium metal as a counter electrode, and glass fiber separator in 1 M NaPF_6_ in a diethylene glycol dimethyl ether (DEGDME) as an electrolyte. Galvanostatic tests with a constant charge/discharge current were conducted WBCS3000S (Wonatech, Seoul, Republic of Korea) in a voltage range of 0.01 to 2.7 V versus Na/Na^+^ at room temperature. Additionally, Cyclic voltammetry (CV) and electrochemical impedance spectra (EIS) measurements were evaluated on a potentiostat (ZIVE SP1, Wonatech, Seoul, Republic of Korea). CV curves were recorded in the voltage range (0.01–2.7 V) at various scan rate. EIS were conducted in the frequency range from 1 MHz to 1 Hz.

## 3. Results and Discussion

### 3.1. Morphology and Composition Analysis

A two-dimensional flake-like VSe_2_ was synthesized through a wet chemical method with various conditions controlled (detailed descriptions can be found in Materials and Methods Sections). Morphologies of VSe_2_@MWCNT and MWCNT were determined using a scanning electron microscope (SEM) and a transmission electron microscope (TEM). The VSe_2_ was well stacked with 1 μm-sized flake-liked nanomaterials, as shown in Figure 1a,b. The crystal lattice spacing of 0.26 nm corresponding to (011) planes showed high crystallinity of VSe_2_ (Figure 1c). The pristine MWCNT in the form of densely distributed nanotubes was entangled with a diameter of about 20 nm (Figure 1d). VSe_2_ was homogenously dispersed throughout the MWCNT. Its shape did not change even after a hybridization process (Figure 1e,f). The MWCNT, which combined well with VSe_2_, could improve electrical conductivity and alleviate volume expansion. The electrode was fabricated without a binder and substrate.

The energy-dispersive spectrometer (EDS) mapping images in Figure 2 show that V and Se were evenly distributed in VSe_2_. Figure 2b,c show the distribution of atomic V and Se and demonstrate the homogeneity of VSe_2_. The stoichiometric V and Se atomic percentages were 31.17% and 68.83%, respectively.

### 3.2. Electrochemical Properties and Sodium-Ion Storage

The crystallinity of VSe_2_@MWCNT and MWCNT was analyzed using X-ray diffraction (XRD), Raman spectrum, and X-ray photoelectron spectroscopy (XPS). Diffraction peaks of VSe_2_@MWCNT were well matched with a combination of JCPDS No.74–1411 and MWCNT (Figure 2a) [38,39,40]. These results showed that they were physically mixed and maintained their compositions even after a composite process. Surface properties of VSe_2_@MWCNT were studied with XPS to confirm binding energy and composition (Figure 3b). Figure 3b displays the XPS V 2p spectrum of VSe_2_@MWCNT. Peaks located at 516.7 eV and 523.9 eV were attributable to V 2p_3/2_ and V 2p_1/2_, respectively [41,42]. As shown in Figure 3c, the Se 3d XPS spectrum exhibited 53.5, 54.3, 55.2 and 55.9 eV (Se 3d_5/2_ and 3d_3/2_). Based on a previous report, trace element Se (0) is highly likely to exist in the final product [43,44,45]. In the XPS C 1s spectrum, peaks at 284.4, 284.9, 285.9 and 289.4 eV corresponded to C-C, C-OH, C=O and C=O-OH, respectively (Figure 3d) [46]. The nitrogen adsorption–desorption measurements were conducted to estimate the surface property of VSe_2_ and VSe_2_@MWCNT (Appendix A). The Brunauer–Emmett–Teller (BET) theory showed a type IV isotherm. VSe_2_@MWCNT exhibited the higher specific surface area of 28.0 m^2^g^−1^ compared to VSe_2_ (16.7 m^2^g^−1^), due to the introduction of MWCNT. The carbon structures of MWCNT and VSe_2_@MWCNT were investigated using Raman spectroscopy. Two representative carbon bands, the E_2g_ vibration mode of graphite layers with sp^2^ carbon (G band at ~1580 cm^−1^) and A_1g_ breathing mode of the sp^2^ bonded carbon near the basal edge corresponding to the structural defects (D band at ~1350 cm^−1^), as well as a 2D band representing the stacked carbon layers, are observed in both MWCNT and VSe_2_@MWCNT spectra [47,48]. The parallel average size of the crystalline sp^2^ carbon clusters (*L_a_*) was calculated from the ratio of the integral Raman intensities of D and G bands. *I_D_/I_G_* used the following Equation (1) [49]:(1)$$\frac{I_{D}}{I_{G}}=\frac{C\left(\lambda \right)}{L_{a}}$$
where *C* (λ) is a constant dependent on the laser wavelength (here, 4.4 for a 514 nm laser). Interestingly, VSe_2_@MWCNT revealed narrower and sharper G and D bands with fewer overlapping disordered carbon peaks at around 1350 and 1580 cm^−1^ related to residual sp^3^ carbons and amorphous sp^2^ carbons, respectively. This showed a more developed carbon sp^2^-hybridized carbon structure compared with that of MWCNT [50]. *I_D_/I_G_* ratio of MWCNTs and VSe_2_@MWCNT were estimated as 1.26 and 0.96, corresponding to 3.50 and 4.57 nm of *L_a_*, respectively.

Electrochemical properties of VSe_2_@MWCNT and MWCNT were evaluated using cyclic voltammetry (CV) and galvanostatic charge/discharge tests with 1 M NaPF_6_ solution with a diethylene glycol dimethyl ether (DEGDME) as an electrolyte. The CV and charge/discharge curve (CD curve) explained the redox reaction of VSe_2_@MWCNT in the potential range of 0.01 to 2.7 V versus Na/Na^+^ (Figure 4a,b). In the first cycle of CV, a reduction peak at 2.03 V was observed, which was related to a sodiation process. In the cathodic sweep process, conversion reaction and formation of solid electrolyte interphase (SEI) occurred at 1.3 V (with a start at about 1.8 V) and 0.3 V peaks [51]. The anodic peaks appeared as a broad 2.41 V peak in the first anodic sweep corresponding to desodiation [52]. These redox peaks were well matched with the plateau at the CD curve. The CV curves changed at each cycle; this was influenced by the formation of SEI layer after the first cycle. As shown in Figure 4c, Nyquist plots of MWCNT and VSe_2_@MWCNT consisted of the depressed semicircle in the high–medium frequency region, as well as the inclined line at low frequency. The EIS data were analyzed and fitted by the proposed equivalent circuit diagram (Appendix A) [53]. The R_sf_ and R_ct_ each showed the SEI layer impedance and charge transfer impedance relevant to the interfacial sodium-ion transfer. The W represented the Warburg impedance related to sodium-ion diffusion at a low frequency. The lower R_ct_ value of VSe_2_@MWCNT (137.6 Ω) than MWCNT (379 Ω) indicated that VSe_2_@MWCNT had outstanding electrical conductivity. To accurately compare the sodium-ion diffusion kinetics of VSe_2_@MWCNT and MWCNT, the sodium diffusion coefficient (*D_Na_*) was calculated using the following Equation (2) [54]:(2)$$D_{Na}=\frac{R^{2}T^{2}}{2A^{2}n^{2}F^{4}C^{2}\sigma^{2}}$$
where *R* is the gas constant; *T* is the absolute temperature; *A* is the surface area of the electrode; *F* is the Faraday constant; and *C* is the concentration of sodium-ion in the electrode. The Warburg factor (σ) is calculated by the slope of the real part resistance and the inverse square root of the angular speed plot in the low-frequency range (Appendix A). The *D_Na_* value of VSe_2_@MWCNT and MWCNT was 6.77 × 10^−19^ cm^2^ s^−1^ and 4.16 × 10^−20^ cm^2^ s^−1^, respectively. These results showed that sodium-ion diffusion of VSe_2_@MWCNT was faster than MWCNT. Figure 4d shows the rate capability results of VSe_2_@MWCNT and MWCNT at different current densities ranging from 0.05 to 2 Ag^−1^ (Figure 4d). The VSe_2_@MWCNT exhibited discharge capacities of 319.6, 274.3, 239.2, 204.6, 186.9, 164.8 and 140.8 mAhg^−1^ at 0.05, 0.1, 0.2, 0.5, 0.8, 1 and 2 Ag^−1^, respectively. A discharge capacity of 252.3 mAhg^−1^ was recovered when the current density was lowered from 2 Ag^−1^ to 0.05 Ag^−1^. The MWCNT exhibited a discharge capacity of 128.3, 123.5, 119.2, 112.5, 107.6, 104.3 and 94.9 mAhg^−1^ at 0.05, 0.1, 0.2, 0.5, 0.8, 1 and 2 Ag^−1^ and recovered the discharge capacity of 127.4 mAhg^−1^ well when the current density returned to 0.05 Ag^−1^. The cycle performance of VSe_2_@MWCNT was conducted at a current density of 0.01 Ag^−1^ (Figure 4e). VSe_2_@MWCNT delivered a discharge capacity of 469.1 mAhg^−1^ after 200 cycles and the Coulombic efficiency reached 99.7%. Coulombic efficiency, that was initially 22.4%, increased to around 99% with cycle progression. To compare the properties of binder-free electrodes for SIBs, the synthesis method and electrochemical performance were summarized in Table 1. Among the free-template metal chalcogenides electrode, VSe_2_@MWCNT showed good sodium-ions storage capacity and Coulombic efficiency. These results showed several advantages of our materials, as follows: (1) MWCNT can enhance intercalation/deintercalation of sodium-ions and alleviate volume expansion during the charge/discharge processes [55]; (2) VSe_2_ has adsorption sites to interact with sodium-ions [32]; (3) non-covalent functionalization of VSe_2_@MWWCNT hybrid affects to electrical conductivity [24].

To study sodium-ion storage and quantitative kinetics of VSe_2_@MWCNT, CV tests were conducted at various scan rates at 0.2 to 2 mVs^−1^ (Figure 5a). With an increasing scan rate, the value of peak current increased proportionally, whereas the shape of the CV did not change. The peak current (mA) and scan rate (mVs^−1^) had a correlation, as shown in Equations (3) and (4). In Equation (4), the value of b (0.5 < *b* < 1) could be calculated based on the slope of the log (peak current)-log (scan rate) plot [58].
(3)$$i\;\left(V\right)=a\nu^{b}$$
(4)$$\log i\;\left(V\right)=b\log\nu +\log a$$

The value of *b* close to 0.5 was a diffusion-controlled reaction (Faradaic process). A value close to 1 indicated a capacitive-controlled reaction (non-Faradaic process) [59]. The *b* value was 0.61 at an oxidation peak of 2.4 V and 0.63 at a reduction peak of 1.3 V (Figure 5b). Electrochemical kinetics at a fixed scan rate could be explained by Equations (5) and (6). The ratio of the capacitive-controlled reaction was given with the value of *k*_1_*ν*, while the diffusion-controlled reaction was given with *k*_2_*ν*^1/2^ [60].
(5)$$i\;\left(V\right)=k_{1}\nu +k_{2}\nu^{1/2}$$
(6)$$\frac{i\;\left(V\right)}{\nu^{1/2}}=k_{1}\nu^{1/2}+k_{2}$$

The obtained proportion of capacitive contribution (brown region) was 85.5% at a scan rate of 1 mVs^−1^ for VSe_2_@MWCNT (Figure 5c). Figure 5d shows the trend of capacitive behavior when the scan rate was increased (proportions of 42.3, 68.9, 85.5 and 98.1% at a scan rate of 0.2, 0.5, 1 and 2 mVs^−1^, respectively). These results indicate that a capacitive behavior-controlled process is better than a diffusion-controlled process. As previously reported, high capacitive contribution enables better cycle performance and plays a pivotal role in good stability and a long lifespan [61,62,63]. Therefore, the two-dimensional flake-like VSe_2_ and the high electrical conductivity of MWCNT can improve the diffusion kinetics of sodium ions.

## 4. Conclusions

In summary, a flake-like VSe_2_ was synthesized with a colloidal method. It was hybridized with MWCNT through vacuum filtration. Such flake-like nanomaterials increased the surface area and made it easier to encounter electrolytes, thus facilitating sodium-ions kinetics. By using MWCNT with excellent flexibility and electrical conductivity, VSe_2_ problems could be solved. Electrodes were prepared without a binder and substrate by hybridization of MWCNT with VSe_2_. Therefore, using Raman spectroscopy, non-covalent functionalized CNT and VSe_2_ composite showed increased sp^2^ carbon structure crystallinity. The hybrid anode exhibited a high level of coulombic efficiency of 99.7 % and a discharge capacity of 469.1 mAhg^−1^, even after 200 cycles. The VSe_2_@MWCNT electrode measured various current densities and showed specific capacities of 319.6, 274.3, 239.2, 204.6, 186.9, 164.8 and 140.8 mAhg^−1^ at 0.05, 0.1 0.2, 0.5, 0.8, 1.0 and 2.0 Ag^−1^. We believe that such binder-free VSe_2_@MWCNT composite films can be easily prepared through the strategy described in this work and be successfully applied as new anode materials in SIBs.

## Figures and Tables

**Figure 1 materials-16-01253-f001:**
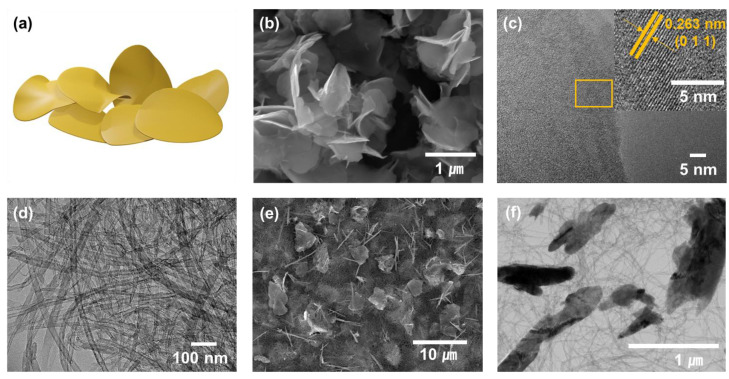
Morphological characterization of the VSe_2_, MWCNT, VSe_2_@MWCNT. (**a**) Illustration of flake-like VSe_2_, (**b**) SEM image of VSe_2_, (**c**) HRTEM of VSe_2_. (**d**) TEM image of MWCNT. (**e**) SEM image, (**f**) TEM image of VSe_2_@MWCNT.

**Figure 2 materials-16-01253-f002:**
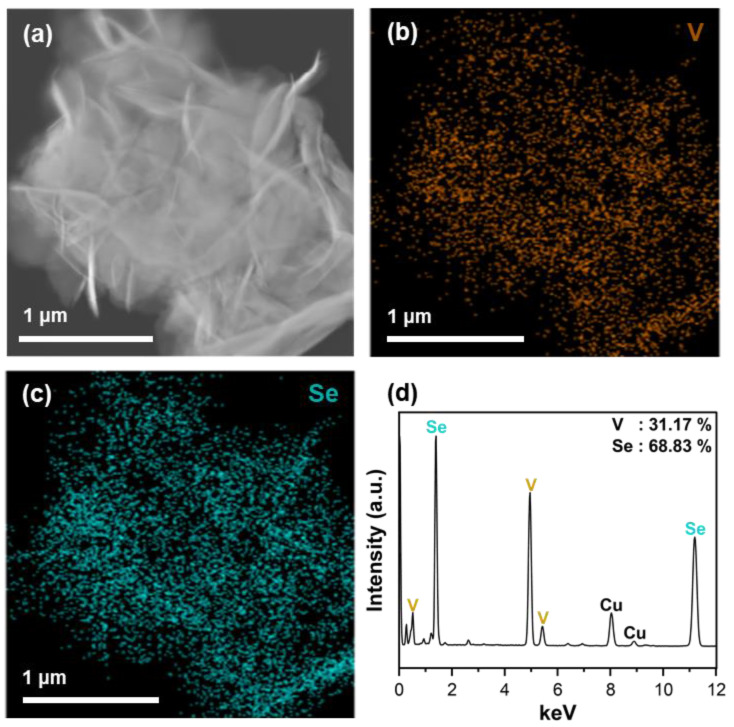
(**a**) TEM image of VSe_2_. Elemental mapping image of (**b**) vanadium, (**c**) selenium. (**d**) EDS spectrum.

**Figure 3 materials-16-01253-f003:**
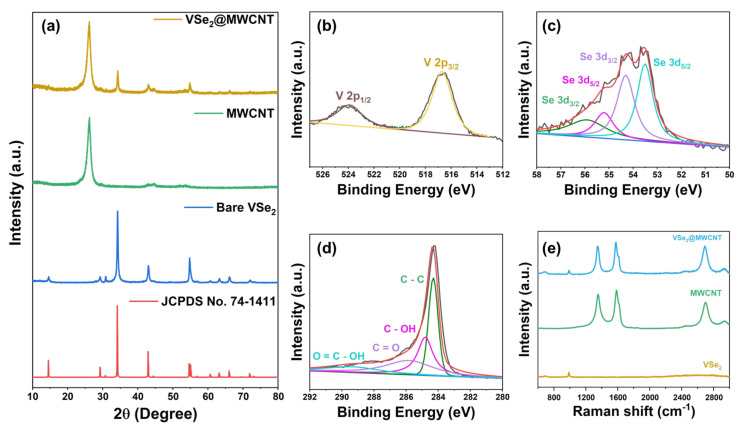
Crystallinity and composition. (**a**) XRD patterns of the VSe_2_, MWCNT, VSe_2_@MWCNT. (**b**) XPS V 2p, (**c**) XPS Se 3d_5/2_ and 3d_3/2_ (**d**) XPS C 1s of the VSe_2_@MWCNT, (**e**) Raman spectrum of the VSe_2_@MWCNT.

**Figure 4 materials-16-01253-f004:**
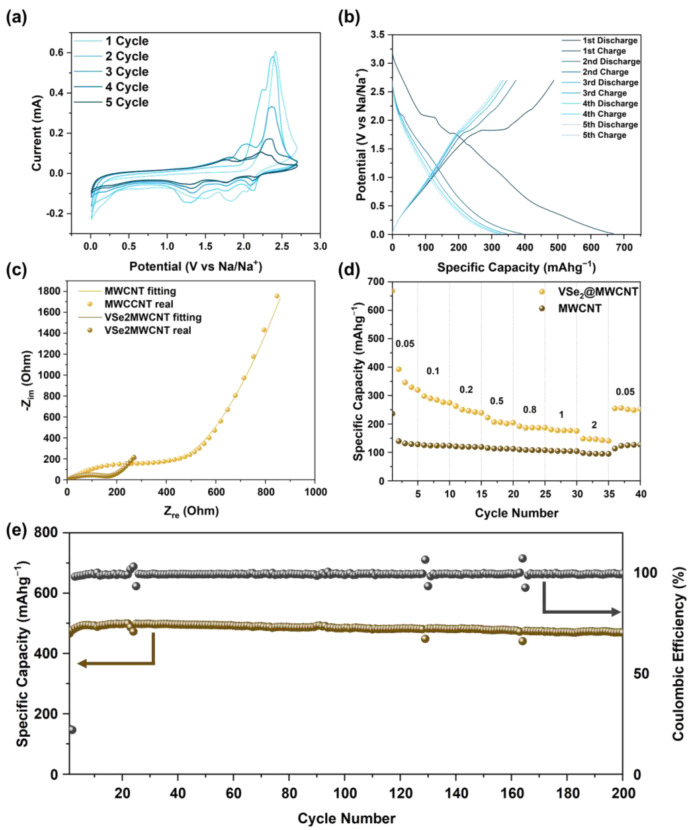
Electrochemical properties over a voltage window between 0.01 to 2.7 V versus Na/Na^+^ (**a**) Cyclic voltammetry at a scan rate of 1 mVs^−1^ of VSe_2_@MWCNT and (**b**) charge/discharge curve at a current density of 0.05 Ag^−1^ of VSe_2_@MWCNT. (**c**) electrochemical impedance spectroscopy diagram and (**d**) rate capabilities at different current densities of MWCNT and VSe_2_@MWCNT. (**e**) cycling performance and coulombic efficiency of VSe_2_@MWCNT at a current density of 0.01 Ag^−1^.

**Figure 5 materials-16-01253-f005:**
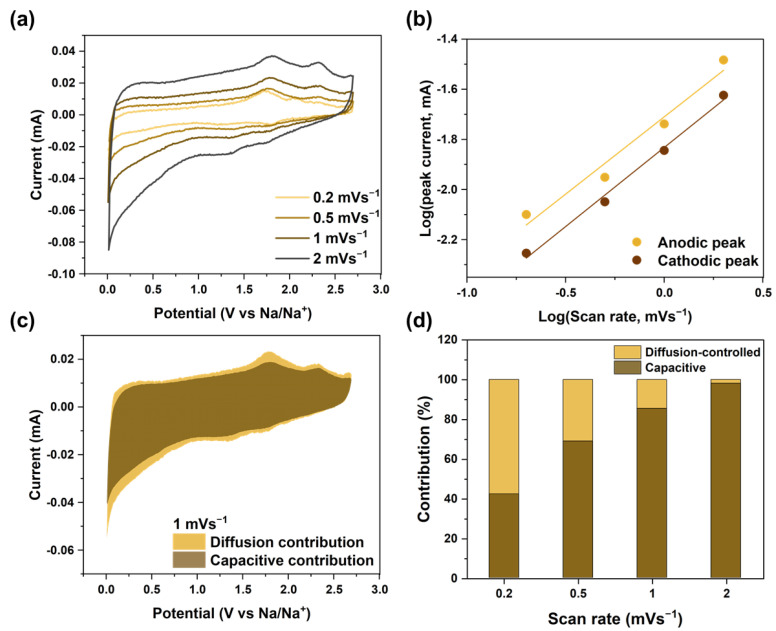
Quantitative capacitive analysis of VSe_2_@MWCNT: (**a**) CV curves at various scan rates from 0.2 to 2 mVs^−1^, (**b**) the relationship between scan rate and peak current, (**c**) the capacity contribution in CV curve at 1 mVs^−1^, (**d**) the proportion of capacitive contributions at various scan rates.

**Table 1 materials-16-01253-t001:** Comparison of electrochemical performance of different kinds of binder-free electrode materials.

Material	Year	Potential	Templates	Synthetic Methods	Electrochemical Performance	Reference
VSe_2_@MWCNT	2023	0.01–2.7 V	Free	Vacuum filtrate	469.1 mAg^−1^ at 10 mAg^−1^ after 200 cycles	This work
FeS@C	2022	0.5–3.0 V	Carbon cloth	Hydrothermal method and carbonization	150 mAhg^−1^ at 12 C after 200 cycles	[11]
VSe_2_/NCNFs	2020	0.01–3.0 V	Carbon fibers	Electrospinning	420.8 mAhg^−1^ at 50 mAg^−1^	[13]
Ultralong Sb_2_Se_3_	2016	0.01–3.0 V	Free	Vacuum filtrate	289 mAhg^−1^ at 100 mAg^−1^ after 50 cycles	[56]
MoS_2_/graphene composite paper	2014	0.0–2.25 V	Free	Vacuum filtrate	218 mAhg^−1^ at 25 mAg^−1^ after 20 cycles	[57]

## Data Availability

The data presented in this study are contained within the article.

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
