# Peer review of "Hybrid Nano Flake-like Vanadium Diselenide Combined on Multi-Walled Carbon Nanotube as a Binder-Free Electrode for Sodium-Ion Batteries"

_materials, 2023, doi:10.3390/ma16031253_

Round 1

Reviewer 1 Report

The article titled “Hybrid Nano Flake-Like VSe2@MWCNT as Binder-free electrode for Sodium-ion Batteries” explores the anode performance of VSe2@MWCNT material in sodium-ion batteries without binders and additives.

VSe2 is a typical transition metal dichalcogenide and has been previously studied in the literature as an anode with different carbon derivatives. In this study, the investigation of the battery performance of VSe2@MWCNT without adding binders and additives to the anode material makes the study attractive.

The manuscript is well-organized and the sections are well-developed. However, the manuscript would benefit from the detailed editing that I mentioned below. I believe that the publication of the manuscript in Materials after some revisions will contribute to sodium ion battery studies.

1.      The introduction part can be improved.  It should be presented more clearly why VSe2 and MWCNT were selected for this study. For example, are these two materials used together to eliminate each other's disadvantages?

2.      In the 2.3. Material Characterization section: Descriptions of XRD, RAMAN, and XPS techniques in the Material Characterization section starting on page 3 line 94, right after these techniques, such as "X-ray diffraction (XRD) were conducted using the D8 Advance A25(Bruker, Billerica, MA, USA)" will be clearer to the reader.

3.      In the 2.4. Electrochemical properties evaluation section: Although it is stated that the substrate is not used in the Abstract section, the author stated that substrate is used in this section. The author should clarify this point. It should also be explained exactly how the electrode was formed. For example, if the electrode was only made with VSe2@MWCNT, was it used in pellet form? What is the diameter of the electrode? How many mg of material is in the electrode on average? It is seen that no information is given about the impedance measurement technique.

4.      In Figure 4 (a) v (b), it is very difficult to distinguish as the colors of the results are very close to each other (some may even be the same). Colors should be chosen in such a way that they can be distinguished from each other.

5.      Scale, label and axis labels in the figures in the text need to be larger in order to be readable. For example, in Figure 3, the labels of XPS peaks cannot be read.

6.      In the Results and Discussion section, it is stated that the Rct’s of the samples are similar. If these constants are calculated, the results should be presented in the manuscript. In addition, in order to talk about these constants, equivalent circuits must be determined. These equivalent circuits should also be presented in the "Results and Discussion" section.

7.      The author stated on page 3 line 127 that “The VSe2 well combined with MWCNT improved electrical conductivity and prevented the volume expansion.” For an argument about conductivity, it would be more accurate to calculate and compare the sodium diffusion coefficients (DNa) of MWCNT and VSe2@MWCNT samples. Also, "prevented" is a pretty assertive word. How the volume expansion was "prevented" should be explained by the author.

8.      The capacity results can be compared with similar studies in the literature. At the end of the Result and Discussion section, adding a table comparing the capacities and operating voltage ranges with reference will increase the quality of the study.

 For these reasons, I recommend a minor revision.

Reviewer 3 Report

Reviewer Comment for Editor/Editor-in-Chief:

The present manuscript presents a study regarding synthesis of nano flake-like VSe2 MWCNT composite and using it as anode material for SIBs.

This manuscript could potentially be suitable for publication, but it needs a minor revision before it could be published.

1.          The title shouldn't include any abbreviations. Please change the manuscript title accordingly.

2.          The abstract is too short and missing the experimental part. Please rewrite and include the experimental part to be more informative.

3.          Also, the introduction s too short. Since the main core of the manuscript is mainly depending on carbon nanotube functionalization (especially non-covalent). Therefore, it is highly recommended to add more explanation and clarification in the introduction section (i.e. one paragraph) discussing this topic including some literature, here are some suggested literatures:

·     Angew. Chem. Inter. Ed., 2002, 41,11, 1853-18593.

·     Appl. Surf. Sci., 2018, 462, 904-912.

·     Chirality. 2020, 32, 3, 345-352.

4.          Line 18 in the abstract, the font type is different. Please modify.

5.     In Figure 1, a) illustration of what? b) SEM image of what? e) SEM image of what?. The authors should write a complete figure caption!

6.     Figures 1e and 1f better to be with the same scale bar for supporting the results.

7.     In page 3, lines 126 and 127; the authors claimed that “The VSe2 well combined with MWCNT improved electrical conductivity and prevented the volume expansion.”. How did the authors prove this claim in this part of the manuscript? No experimental evidence?

8.     In Figure 2d, the peak around 11 keV should be assigned as it may affect the wt% of V and Se.

9.          Since there some typing and grammatical errors appeared in this manuscript, therefore the authors should check the manuscript very carefully and correct all possible typing and grammatical errors. For example, page 4, line 136 “Figure2b and 2c describes!” and so on.

10.       All the references should be revised very well. The references must be uniformly formatted.  Many references are written with many mistakes. See ref. 1, 2, 4, 5, 6, 8 and 9 for example. Also, ref. 3 is completely missed.

Round 2

Reviewer 2 Report

The authors have improved the revised version and can be accepted for publication.

Reviewer 3 Report

The authors have addressed all the comments properly, therefore, the manuscript is suitable for publication in Materials in the current form